# MAJL: A Model-Agnostic Joint Learning Framework for Music Source Separation and Pitch Estimation

## ABSTRACT

Music source separation and pitch estimation are two vital tasks in music information retrieval. Typically, the input of pitch estimation is obtained from the output of music source separation. Therefore, existing methods have tried to perform these two tasks simultaneously, so as to leverage the mutually beneficial relationship between both tasks. However, these methods still face two critical challenges that limit the improvement of both tasks: the lack of labeled data and joint learning optimization. To address these challenges, we propose a Model-Agnostic Joint Learning (MAJL) framework for both tasks. MAJL is a generic framework and can use variant models for each task. It includes a two-stage training method and a dynamic weighting method named *Dynamic Weights on Hard Samples* (DWHS), which addresses the lack of labeled data and joint learning optimization, respectively. Experimental results on public music datasets show that MAJL outperforms state-of-the-art methods on both tasks, with significant improvements of 0.92 in Signal-to-Distortion Ratio (SDR) for music source separation and 2.71% in Raw Pitch Accuracy (RPA) for pitch estimation. Furthermore, comprehensive studies not only validate the effectiveness of each component of MAJL, but also indicate the great generality of MAJL in adapting to different model architectures.

## CCS CONCEPTS

• **Applied computing** → **Sound and music computing**.

## KEYWORDS

music source separation, pitch estimation, joint learning

## 1 INTRODUCTION

The digital music industry has been growing rapidly in the last few years due to the mass publication of music through smartphone apps, enabling hundreds of millions of people to access a song via large music platforms. This has created huge music streaming companies such as Spotify (worth $37B) and QQ Music (worth $10B). The digital music industry in the US has a market of close to $10B in 2022 and has been growing at more than 10% in each of the last five years [39] and that in China has a market of $5B and has been growing at 30~50% in each of the last five years [26].

Music Information Retrieval (MIR) is a pivotal research domain that supports the functionality of large music platforms. Within

MIR, music source separation (MSS) and pitch estimation (PE) emerge as critical tasks with far-reaching implications. MSS facilitates several downstream tasks, such as lyrics extraction [15] and music transcription [2], accentuating its pivotal role. PE has great significance for various MIR applications, including content-based music recommendation and query/search by singing [60]. Notably, real-world musical compositions are often mixture music, typically as .wav or .mp3 files, which do not inherently provide the pitch information of music data. To extract target sources along with their corresponding pitches, simultaneous execution of both MSS and PE tasks becomes imperative.

The MSS task entails generating isolated stems for vocals, bass, or drums from raw audio or spectrograms of mixture music, as shown in the top row of Figure 1(a). The PE task involves extracting fundamental frequencies ($f0$) from clean music audio or spectrograms, as shown in the second row of Figure 1(a). Following previous studies [18, 32, 61], we default to using pitch estimation to refer to single pitch estimation unless explicitly stated otherwise in this paper. It is important to note that the clean music in the PE task corresponds to the predicted source in the MSS task because both are music data from a specific instrument.

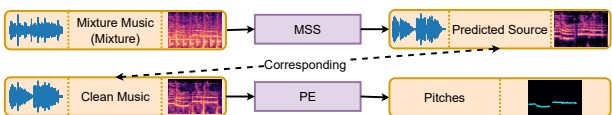

(a) The flow chart of pipeline methods for MSS and PE.

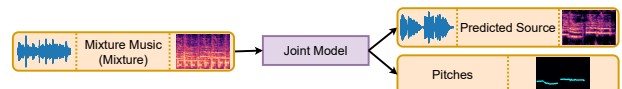

(b) The flow chart of joint learning methods for MSS and PE.

**Figure 1: Existing methods for MSS and PE.**

Music source separation and pitch estimation are closely related in music information retrieval, where the input of pitch estimation is typically obtained from the output of music source separation, as shown in Figure 1(a). Consequently, various studies have aimed to simultaneously tackle these two tasks, leveraging on their mutually beneficial relationship. One line of studies (e.g., DNN+UPDUDP [15] and HR-ED [16]) uses *pipeline methods*, as shown in Figure 1(a). However, these methods train MSS and PE models independently on different music datasets, leading to a mismatch between the data distributions at training and testing time. This mismatch limits the improvement of pitch estimation from mixture music. Another line of studies (e.g., HWJH [23], HS-W$_p$ [44], and S→P→S→P [28]) employs *joint learning methods*, as shown in Figure 1(b). Although these joint models combine MSS and PE tasks by summing the respective losses, the distinct objectives of each task pose a challenge to achieve simultaneous improvements. Moreover, these models are often designed for specific tasks, lacking scalability, thus limiting

their ability to improve performance even when better MSS or PE models become available. Although the above studies have tried to perform MSS and PE tasks simultaneously, there are still two major challenges have to be addressed as described below.

**Challenge 1: Lack of labeled data.** The field of music information retrieval suffers from a large scarcity of annotated datasets, particularly due to the laborious and professional nature of obtaining target sources and corresponding pitches, even for experienced musicians. As a result, only a small amount of music datasets (e.g., MIR-1K [22] and MedleyDB[1] [4]), offer both target sources and corresponding pitches. But the total duration of MIR-1K is only 2.25 h, and the total duration of MedleyDB is 3.21 h. We call such kind of dataset as _fully-labeled dataset_. In contrast, datasets specific to MSS or PE are much larger than the fully-labeled dataset. For example, MUSDB18 [50] is a music source separation dataset and the total duration of MUSDB18 is about 5 times as that of MIR-1K. While MIR_ST500 [58] is a dataset for pitch estimation from mixture music and the total duration of MIR_ST500 is about 14 times as that of MIR-1K. We call such kind of datasets as _single-labeled dataset_. The small amount of fully-labeled datasets prevent models from effectively learning the relationship between MSS and PE tasks.

**Challenge 2: Joint learning optimization.** Existing joint learning methods (e.g., HS-W$_p$ [44] and S→P→S→P [28]) design joint models and perform joint learning by simply summing up the losses of both tasks. However, these methods do not address the following problems: (i) _Error propagation._ Due to the cascade relationship between MSS and PE, poor predictions of music source separation can lead to poor pitch estimation results. This error propagation problem is critical in joint learning of both tasks. (ii) _Misalignment between different objectives._ The objectives of MSS and PE differ from each other. As a result, simply adding the losses of these two tasks cannot guarantee simultaneous improvements in MSS and PE. We believe that existing joint learning methods failed to align different objectives because they treated all samples equally. These two problems make the joint learning of MSS and PE challenging.

To address these challenges, we propose a Model-Agnostic Joint Learning (MAJL) framework for music source separation and pitch estimation. Our framework can adopt existing MSS and PE models and improve the performance of both tasks when better models become available. MAJL contains a two-stage training method and the Dynamic Weights on Hard Samples (DWHS), which is designed to address Challenge 1 and Challenge 2, respectively.

To address Challenge 1, we design a two-stage training method to leverage large single-labeled datasets. This method comprises an initialization stage (Stage I) and a semi-supervised training stage (Stage II). In Stage I, the model is trained using the available fully-labeled data, then utilizing this trained model to generate pseudo labels and corresponding confidence values for the single-labeled data. The confidence value reflects the quality of pseudo labels. In Stage II, we retrain the model from scratch, using the music data that consists of fully-labeled data, single-labeled data, and pseudo labels generated during Stage I. Additionally, a threshold-based filter is applied to exclude low-confidence single-labeled data, ensuring the

---

[1]The MedleyDB dataset comprises a total duration of 5.56 h and encompasses a wide range of musical instruments. Within this dataset, a subset of 3.21 h contains clean vocals along with their corresponding pitches. Notably, the duration of recordings for each other instrument is less than 1 h. Thus, we focus on clean vocals here.

quality of pseudo labels. This two-stage training method effectively leverages extensive single-labeled datasets and high-quality pseudo labels, effectively addressing the lack of labeled data.

For Challenge 2, we design a dynamic weighting method called Dynamic Weights on Hard Samples (DWHS), to solve the problems of error propagation and misalignment between different objectives in joint learning. Addressing the error propagation problem entails identifying the module making poor predictions in our framework. Furthermore, to solve the problem of misalignment, we need to identify hard samples for both the MSS and PE tasks. The DWHS entails extracting pitch results from target and predicted sources, termed _target_source2Pitch_ and _predicted_source2Pitch_, respectively. By comparing these pitch results, we not only identify modules producing poor predictions but also identify hard samples for both tasks. This allows us to allocate appropriate weights to hard samples across tasks, thereby mitigating error propagation and aligning different objectives within joint learning.

In this paper, our contributions are summarized as follows:

- We propose a Model-Agnostic Joint Learning (MAJL) framework for music source separation and pitch estimation. MAJL is a generic framework, which can further improve the performance of both tasks when better MSS or PE models are available. Moreover, our framework outperforms previous methods, leading to significant improvements in the performance of both tasks.
- We design a two-stage training method to solve the lack of labeled data. The two-stage training method combines music source separation and pitch estimation tasks at the data aspect. By leveraging large single-labeled datasets, our method extends fully-labeled data, allowing the model to better learn the relationship between both tasks.
- To address the challenge of joint learning optimization, we design a novel dynamic weighting method, named Dynamic Weights on Hard Samples (DWHS). The DWHS can handle error propagation and misalignment between different objectives by identifying hard samples and setting appropriate weights for these samples.

## 2 RELATED WORK

**Music Source Separation (MSS)** is a crucial task in MIR, aiming to isolate individual sources from mixture music. Many deep learning methods have been proposed for MSS, generally categorized into common models and side-information informed models.

Common models operate solely on hidden features extracted from the time or frequency domains. For example, Spleeter [21], U-Net [29], CWS-PResUNet [40] and ResUNetDecouple+ [36] predict target sources using frequency domain features. In contrast, Demucs [10], Wave-U-Net [57] and its follow-ups [41, 47] leverage time domain features. Other methods such as KUIELAB-MDX-Net [33], Hybrid Demucs [9] and HT Demucs [53] fuse both domain features to enhance the performance of MSS. In contrast, the side information informed models use additional information, such as lyrics, pitches, or spatial information to improve MSS. For example, JOINT3 [54] employs phoneme-level lyrics alignment, while Soundprism [12, 14] and SPAIN-NET [48] leverage pitches and spatial

information, respectively. These methods exclusively address the MSS task, which do not support simultaneous PE task.

**Pitch Estimation (PE)** is a fundamental task in MIR, aimed at extracting fundamental frequency ($f0$). PE can be broadly classified into two sub-tasks: PE from clean music and PE from mixture music.

For clean music, existing methods encompass heuristic-based and data-driven methods. Heuristic-based methods like ACF [13], YIN [8], SWIPE [5], and pYIN [42] leverage candidate-generating functions to predict pitches. Conversely, data-driven methods, including CREPE [32], DeepF0 [56], and HARMOF0 [61], rely on supervised training of models for PE. While these methods achieve accurate pitch results from clean music, their performance is constrained when applied to mixture music due to the presence of other existing sources. For mixture music, existing methods comprise pipeline and end-to-end methods. Pipeline methods involve utilizing MSS models (e.g., Spleeter [21] and U-Net [29]) to extract target sources from the mixture music, and then using PE models to predict corresponding pitches. However, a mismatch between the data distributions at training and testing times often limits the performance of PE from mixture music. End-to-end methods (e.g., DSM-HCQT [3], CNN-Raw [11], and JDC [37]) are designed to directly predict pitches from mixture music. Nevertheless, these methods encounter performance limitations as a result of the presence of other sources in mixture music.

## 3 PROBLEM FORMULATION

In this section, we formulate the problems of Music Source Separation (MSS) and Pitch Estimation (PE), along with a previously proposed naive joint learning method referred to as the Joint Cascade Framework (JCF) in this paper.

**Music Source Separation (MSS).** The task of MSS is to extract a target source from mixture music signals. The mixture music signals can be represented as either the raw audio waveform $x$ or its corresponding spectrogram $X_{T \times F}$, where $T$ is the number of audio frames and $F$ is the number of frequency bins. It should be noted that the spectrogram is computed using the short-time Fourier transform (STFT) as a feature representation of the original music signal. The output of MSS is the target source $s$, and the spectrogram of target source is represented as $S_{T \times F}$. Thus, the MSS task can be formulated as $\mathcal{F}_{mss} : x(X_{T \times F}) \rightarrow s$.

**Pitch Estimation (PE).** The task of PE aims to estimate the pitch sequence of clean music from a raw audio waveform or its spectrogram representation. Following previous studies (e.g., 360 in CREPE [32] and 352 in HARMOF0 [61]), each pitch is typically represented as a $N$-dimensional one-hot vector $y$. As a result, the output of PE is a sequence of pitch vectors $Y_{T \times N}$, where $N$ is the number of pitch values. Furthermore, the input of PE is typically obtained from the output of MSS. Thus, the PE task can be formulated as $\mathcal{F}_{pe} : s(S_{T \times F}) \rightarrow Y_{T \times N}$.

**Joint Cascade Framework (JCF).** The JCF is designed to leverage the cascade relationship between MSS and PE, enabling joint learning of both tasks. It (cf. Figure 3) comprises a Music Source Separation Module (MSS Module) and a Pitch Estimation Module (PE Module). Firstly, the features of mixture music are input into the MSS Module to obtain predicted sources. Then, the PE Module extracts corresponding pitches from the predicted sources.

In line with previous studies [29, 32, 36, 59, 61], the training of JCF involves using the Mean Absolute Error (MAE) loss for MSS and the Binary Cross Entropy (BCE) loss for PE, respectively. Therefore, the loss function for MSS is defined as:

$$\mathcal{L}_{mss}(s, \hat{s}) = \sum_{i=0}^{L} |s_i - \hat{s}_i| \qquad (1)$$

where $s$ is the target sources, $\hat{s}$ is the predicted sources and $L$ is the length of mixture music. And the loss function for PE is defined as:

$$\mathcal{L}_{pe}(y, \hat{y}) = -\sum_{i=0}^{N} (y_i \log \hat{y}_i + (1 - y_i) \log(1 - \hat{y}_i)) \qquad (2)$$

where $N$ is the number of pitch values, $y$ is the ground truth of pitch results and $\hat{y}$ is the predicted pitch value. Thus, the total loss for naive joint learning of MSS and PE is:

$$\mathcal{L}_{total} = \mathcal{L}_{mss} + \mathcal{L}_{pe} \qquad (3)$$

The JCF is unable to solve the two challenges mentioned in Section 1. Therefore, we propose our Model-Agnostic Joint Learning (MAJL) framework for both tasks, building upon and extending the JCF.

## 4 METHOD

As shown in Figure 2, the Model-Agnostic Joint Learning (MAJL) framework contains two important components: two-stage training method and Dynamic Weights on Hard Samples (DWHS). Besides, the music source separation module (MSS Module) and pitch estimation module (PE Module) within MAJL can be easily replaced with existing MSS and PE models.

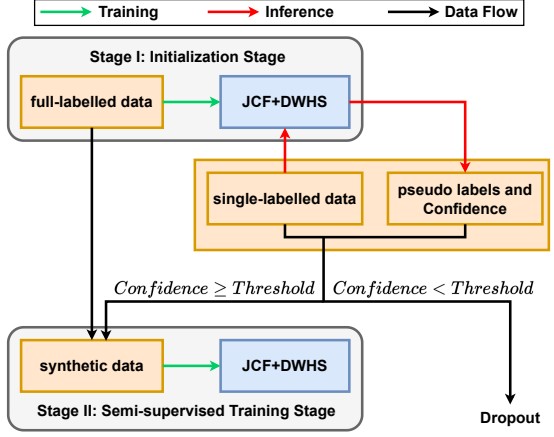

**Figure 2: The overall structure of Model-Agnostic Joint Learning (MAJL) framework. Details of JCF and DWHS are shown in Figure 3. The synthetic data contains fully-labeled data, single-labeled data with generated pseudo labels.**

## 4.1 Two-Stage Training Method

To address the limited availability of fully-labeled datasets and leverage large single-labeled datasets, we design a two-stage training method within our framework. This method comprises an initialization stage (Stage I) and a semi-supervised training stage (Stage II), as shown in Figure 2.

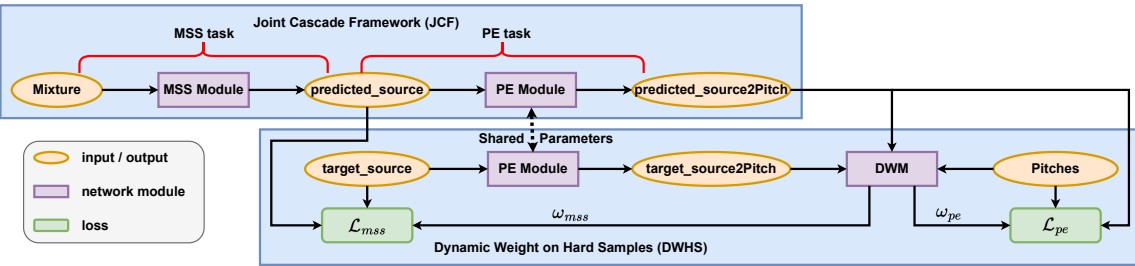

**Figure 3: Details of JCF and DWHS. The MSS Module and the PE Module used existing MSS and PE models respectively. The details of Dynamic Weight Module (DWM) are shown in Section 4.2.2.**

**Initialization Stage (Stage I):** During Stage I, our framework is trained using fully-labeled music data (e.g., MIR-1K or MedleyDB). Then the trained framework is employed to generate pseudo labels for target sources or corresponding pitches. Additionally, we compute confidence values for each pitch result and each frame of target sources. For predicted pitches, a pitch value is considered present when $max(\hat{y}) \geq 0.5$; otherwise, a pitch value is considered absent. Then the confidence value (confi) is defined as:

$$\text{confi} = \begin{cases} max(\hat{y}) & max(\hat{y}) \geq 0.5 \\ 1 - max(\hat{y}) & max(\hat{y}) < 0.5 \end{cases} \quad (4)$$

It should be noted that this confidence value can also be applied to predicted sources due to the inherent cascade relationship between music source separation and pitch estimation.

To maintain a consistent format for confidence values across different dataset types (fully-labeled and single-labeled datasets), we set the confidence values of true labels to 1. Therefore, for fully-labeled datasets (e.g., MIR-1K and MedleyDB), the confidence values of MSS ($\text{confi}_{mss}$) and PE ($\text{confi}_{pe}$) are defined as:

$$\text{confi}_{mss} = 1 \quad \text{confi}_{pe} = 1 \quad (5)$$

Then, for MSS datasets (e.g., MUSDB18), which belongs to single-labeled datasets, the confidence values of MSS ($\text{confi}_{mss}$) and PE ($\text{confi}_{pe}$) are defined as:

$$\text{confi}_{mss} = 1 \quad \text{confi}_{pe} = \text{confi} \quad (6)$$

While for PE datasets (e.g., MIR_ST500), which also belongs to single-labeled datasets, the confidence values of MSS ($\text{confi}_{mss}$) and PE ($\text{confi}_{pe}$) are defined as:

$$\text{confi}_{mss} = \text{confi} \quad \text{confi}_{pe} = 1 \quad (7)$$

**Semi-supervised Training Stage (Stage II):** After the initialization stage, we obtain pseudo labels and confidence values from single-labeled datasets. We then combine fully-labeled music data, single-labeled music data, and pseudo-labels to create a synthetic dataset. To filter pseudo-labels from single-labeled music data, we set a threshold ($th$). The detailed filtering process involves applying weights for MSS and PE in the loss computation. Subsequently, we retrain our framework from scratch using the synthetic dataset. Thus, the loss function of stage II is written as:

$$\mathcal{L}_{total} = \text{confi}_{mss} \times \mathcal{L}_{mss} + \text{confi}_{pe} \times \mathcal{L}_{pe} \quad (8)$$

Here, $\text{confi}_{mss}$ equals 1 if $\text{confi}_{mss} \geq th$, and 0 otherwise. $\text{confi}_{pe}$ is calculated using the same way as $\text{confi}_{mss}$.

## 4.2 Dynamic Weights on Hard Samples (DWHS)

### 4.2.1 Analysis of Different Cases in DWHS.
The naive joint learning method can not ensure simultaneous improvements in both tasks due to the problems of error propagation and misalignment between distinct objectives. To address these problems concurrently, we should identify hard samples and assign appropriate weights to each sample in both tasks. This can be achieved by comparing the predicted pitches from target sources with those from predicted sources (c.f. Figure 3). By this comparison, we can determine which module is delivering poor predictions and identify samples that are hard for either MSS or PE. A comprehensive analysis of different cases arising from this comparison is summarized in Table 1. Detailed explanations of each case in Table 1 are provided as follows.

For _Case 1_, both the predicted pitches from predicted sources and those from target sources are correct. This result indicates that there is no issue with the MSS Module, the PE Module or the quality of data. For _Case 2_, the predicted pitches from predicted sources are correct, while those from target sources are incorrect. This result indicates the presence of noisy pitch labels in the data. To mitigate the impact of noisy labels, the weights of such samples for PE should be within the range of 0 to 1. For _Case 3_, the predicted pitches from predicted sources are incorrect, while those from target sources are correct. This result indicates that the predicted sources are quite different from the original target sources, making the data hard for the MSS. To emphasize learning on hard samples, the weights assigned to these samples in the MSS should be greater than 1. For _Case 4_, both the predicted pitches from target sources and those from predicted sources are incorrect. This result highlights poor predictions by the PE Module, indicating that the music data is hard for the PE. To emphasize learning on hard samples, the weights assigned to these samples in the PE should be greater than 1.

### 4.2.2 Module of DWHS.
By leveraging the above analysis, we can assign different weights to each sample based on identified cases, thereby aligning the focus of two tasks during joint learning. The most direct method involves setting different weights for different cases as outlined in Table 1. However, this approach incurs high training costs due to the difficulty of manually determining proper weights for each case. Therefore, we introduce the DWHS, which automatically extracts the appropriate weights for different cases.

The DWHS utilizes a simple network called the Dynamic Weight Module (DWM) to determine dynamic weights for the MSS and PE tasks under different cases. As illustrated in Figure 4, the inputs for the DWM consist of _predicted_source2Pitch_, _Pitches_, and _target_source2Pitch_ from Figure 3, maintaining the same format as

**Table 1: Analysis for different cases in DWHS and the weights that should be set for different cases by the DWHS.**

| Case | predicted_source2Pitch | target_source2Pitch | Analysis | | | The Weights Should be Assigned | |
|------|------------------------|---------------------|------------|-----------|------|---------------------------|-------------------------|
| | | | MSS Module | PE Module | Data | $\omega_{mss}$ | $\omega_{pe}$ |
| 1 | Correct | Correct | ✓ | ✓ | ✓ | $\omega_{mss} = 1$ | $\omega_{pe} = 1$ |
| 2 | Correct | Incorrect | ✓ | ✓ | ✗ | $\omega_{mss} = 1$ | $0 \leq \omega_{pe} < 1$ |
| 3 | Incorrect | Correct | ✗ | ✓ | ✓ | $\omega_{mss} \geq 1$ | $\omega_{pe} = 1$ |
| 4 | Incorrect | Incorrect | ✓ | ✗ | ✓ | $\omega_{mss} = 1$ | $\omega_{pe} \geq 1$ |

$Y_{T \times N}$. These inputs are concatenated and passed through two CNN layers with ReLU activation, configured as depicted in Figure 4 with $3 \times 3$ kernels. Following the CNN layers, there is a flatten layer, a fully connected layer with ReLU activation, and finally, a fully connected layer with sigmoid activation that generates dynamic weights for both tasks. This process enables dynamic assignment of weights without the need for manual specification of specific weights. Specifically, the weights ($\omega_{mss}$ and $\omega_{pe}$) corresponding to different cases outlined in Table 1 are automatically extracted by the Dynamic Weight Module (DWM) of the DWHS.

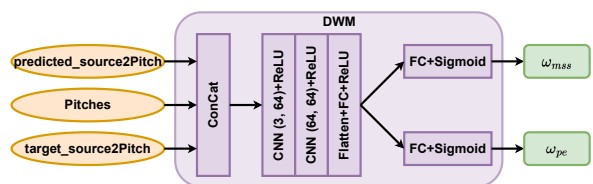

**Figure 4: The model structure of dynamic weight module (DWM) in Dynamic Weights on Hard Samples (DWHS).**

*4.2.3 Loss of DWHS.* To ensure the weights extracted by DWM for different cases align with the specified weights shown in Table 1, we design the loss function for the DWM of DWHS to address four specific cases. For *Case 1*, the Mean Absolute Error (MAE) loss is employed to ensure the predicted weights are around 1. Then the loss function for DWM of DWHS in Case 1 is defined as:

$$\mathcal{L}_{dwhs\_1} = |\omega_{mss} - 1| + |\omega_{pe} - 1| \qquad (9)$$

For *Case 2*, the MAE loss is utilized to ensure the predicted weights of MSS are around 1. For the predicted weights of PE, the Bayesian Personalized Ranking (BPR) loss [52] is employed to ensure lower weights for noisy music data. Then the loss function for DWM of DWHS in Case 2 is defined as:

$$\mathcal{L}_{dwhs\_2} = |\omega_{mss} - 1| - \ln \sigma(1 - \omega_{pe}) \qquad (10)$$

For *Case 3*, the BPR loss is used to ensure hard samples for MSS receive higher weights. Simultaneously, the MAE loss is used to ensure the predicted weights of PE are around 1. Then the loss function for DWM of DWHS in Case 3 is defined as:

$$\mathcal{L}_{dwhs\_3} = -\ln \sigma(\omega_{mss} - 1) + |\omega_{pe} - 1| \qquad (11)$$

For *Case 4*, the MAE loss is employed to ensure the predicted weights of MSS are around 1. Additionally, the BPR loss is used to ensure hard samples for PE receive higher weights. Then the loss function for DWM of DWHS in Case 4 is defined as:

$$\mathcal{L}_{dwhs\_4} = |\omega_{mss} - 1| - \ln \sigma(\omega_{pe} - 1) \qquad (12)$$

Thus, the total loss of DWHS is written as:

$$\mathcal{L}_{dwhs} = \mathcal{L}_{dwhs\_1} + \mathcal{L}_{dwhs\_2} + \mathcal{L}_{dwhs\_3} + \mathcal{L}_{dwhs\_4} \qquad (13)$$

It is important to note that if there is no music data belonging to a specific case, the loss function for that case is automatically set to 0. For example, if there are no music data belonging to Case 1, then the $\mathcal{L}_{dwhs\_1}$ becomes 0.

Thus, with the DWHS, the loss function of stage I is written as:

$$\mathcal{L}_{total} = \omega_{mss} \times \mathcal{L}_{mss} + \omega_{pe} \times \mathcal{L}_{pe} + \mathcal{L}_{dwhs} \qquad (14)$$

And the loss function of stage II is written as:

$$\mathcal{L}_{total} = \text{confi}_{mss} \times \omega_{mss} \times \mathcal{L}_{mss} + \text{confi}_{pe} \times \omega_{pe} \times \mathcal{L}_{pe} + \mathcal{L}_{dwhs} \quad (15)$$

where $\text{confi}_{mss}$ and $\text{confi}_{pe}$ are the same as those in Eq. 8.

The above two-stage training method and Dynamic Weights on Hard Samples are two important components of MAJL aimed at addressing the **lack of labeled data (Challenge 1)** and the **joint learning of optimization (Challenge 2)**, respectively. The effectiveness of MAJL and each component is evaluated in Section 6.

## 5 EXPERIMENTAL SETUP

**Datasets.** We evaluate our framework using four public datasets: MIR-1K [22], MedleyDB [4], MIR_ST500 [58], and MUSDB18 [50]. MIR-1K and MedleyDB provide both mixture and clean vocal tracks, along with pitch labels for vocal parts, making them fully-labeled datasets. In contrast, MIR_ST500 and MUSDB18 provide PE and MSS labels, respectively, classifying them as single-labeled datasets. It should be noted that all pitch labels are transformed into frequency bins represented in Hz format the same as MIR-1K.

**Evaluation Metrics.** Following previous studies, we use four metrics to evaluate our framework. For MSS, we use Signal-to-Distortion Ratio (SDR) [36], Global Normalized Signal-to-Distortion Ratio (GNSDR) [22] to evaluate the quality of predicted sources. A higher SDR or a higher GNSDR indicates better separation results, and vice versa. For PE, we use Raw Pitch Accuracy (RPA) and Raw Chroma Accuracy (RCA) [32] to evaluate the accuracy of predicted pitches.

**Implementation Details.** The raw audio is sampled at 16kHz and then transformed into spectrograms using the short-time Fourier transform (STFT) with a Hann window size of 2048 and a hop length of 320 (20ms). During the training of MAJL, we use a batch size of 16 and the Adam optimizer [34]. The learning rate is initialized to 0.001 and then reduced by 0.98 of the previous learning rate every 10 epochs. For MIR-1K [22] and MedleyDB [4] datasets, we randomly split these datasets into training (80%) and testing (20%) sets. The splitting way for MIR_ST500 and MUSDB18 datasets is introduced in [58] and [50], respectively. During experiments, we only consider the target source of vocals due to the lack of fully-labeled data from other sources such as bass and drums.

## 6 EXPERIMENTAL RESULTS

In this section, we present the experimental results of our framework to show the superiority of MAJL. We firstly compare MAJL

Table 2: Performance comparison for MSS and PE tasks on the MIR-1K [22] and MedleyDB [4] datasets. "Extra" indicates the extra single-labeled music data used at the training time. "Both" means MUSDB18 and MIR_ST500.

| Methods | Extra | MIR-1K | | | | MedleyDB | | | |
| --- | --- | --- | --- | --- | --- | --- | --- | --- | --- |
| | | MSS | | PE (%) | | MSS | | PE (%) | |
| | | SDR | GNSDR | RPA | RCA | SDR | GNSDR | RPA | RCA |
| *End-to-End Methods* | | | | | | | | | |
| CNN-Raw [11] | ✗ | —— | —— | 81.70 | 90.90 | —— | —— | 64.33 | 66.42 |
| JDC [37] | ✗ | —— | —— | 87.47 | 88.00 | —— | —— | 69.58 | 77.15 |
| *Pipeline Methods w/i CREPE* [32] | | | | | | | | | |
| U-Net [29] | ✗ | 11.43 | 8.48 | 89.28 | 90.41 | 5.06 | 8.75 | 72.65 | 74.98 |
| ResUNetDecouple+ [36] | ✗ | 12.06 | 9.13 | 91.40 | 92.07 | 5.54 | 10.31 | 74.62 | 76.29 |
| *Pipeline Methods w/i HARMOF0* [61] | | | | | | | | | |
| U-Net [29] | ✗ | 11.43 | 8.48 | 87.95 | 88.57 | 5.06 | 8.75 | 71.24 | 73.78 |
| ResUNetDecouple+ [36] | ✗ | 12.06 | 9.13 | 90.21 | 90.61 | 5.54 | 10.31 | 73.38 | 75.90 |
| *Joint Learning Methods* | | | | | | | | | |
| HS-$W_p$ [44] | ✗ | 9.80 | 6.87 | 85.04 | 85.32 | 4.32 | 7.56 | 68.44 | 70.03 |
| S→P→S→P [28] | ✗ | 11.70 | 8.72 | 86.62 | 86.94 | 5.14 | 9.00 | 70.83 | 73.79 |
| MAJL-Stage I | ✗ | 12.33 | 9.36 | 93.17 | 93.65 | 6.04 | 11.12 | 76.07 | 78.28 |
| MAJL | MUSDB18 | 12.81 | 9.86 | 93.38 | 93.88 | 6.91 | 11.87 | 76.91 | 79.43 |
| MAJL | MIR_ST500 | 12.55 | 9.59 | 93.67 | 94.08 | 6.39 | 11.43 | 77.78 | 80.11 |
| MAJL | Both | **12.98** | **9.99** | **94.11** | **94.38** | **7.18** | **12.14** | **78.38** | **83.21** |

with baselines on different datasets in Section 6.1. Then we explore the generality of MAJL through an investigation of its various modules in Section 6.2. Following these experiments, we visualize and analyze the weights of DWHS to understand the effectiveness of DWHS in Section 6.3. Finally, a study is conducted to investigate the threshold ($th$) used in the two-stage training method in Section 6.4.

## 6.1 Overall Performance

In this experiment, we conduct a comprehensive comparison of MAJL with several baselines, encompassing End-to-End methods, pipeline methods, and joint learning methods. We evaluate the performance of MAJL on both fully-labeled datasets and single-labeled datasets. These experimental results not only demonstrate the effectiveness of MAJL in joint learning of both tasks, but also highlight its superiority in either the MSS task or the PE task.

*6.1.1 Results on Fully-labeled Dataset.* To show the superiority of our framework, we perform a comparison with several baselines. For the MSS and PE modules, we choose ResUNetDecouple+ and CREPE, respectively, since they achieve the best performance among all other MSS and PE modules, as shown in Table 5.

Our framework shows the effectiveness in both tasks, achieving state-of-the-art performance for both tasks, as summarized in Table 2. **(i):** Compared to the End-to-End methods, our framework tackles both tasks simultaneously, leading to significant improvement in the PE task. **(ii):** Moreover, our framework outperforms existing joint learning and pipeline methods. Specifically, MAJL-Stage I outperforms the previous best model (Pipeline method with ResUNetDecouple+ and CREPE) by 0.27 in SDR and 1.77% in RPA on the MIR-1K dataset. These results demonstrate that our DWHS is effective in enhancing the performance of both tasks. **(iii):** Furthermore, by incorporating additional single-labeled music data, the performance of both tasks is further improved. For example, MAJL demonstrates a 0.65 improvement in SDR and 0.94% in RPA over MAJL-Stage I when incorporating these additional datasets

(MUSDB18 and MIR_ST500) on the MIR-1K dataset, validating the effectiveness of our two-stage training method. **(iv):** Besides, the experimental results on the MedleyDB dataset is similar with those on the MIR-1K dataset, showing the effectiveness of MAJL. Further analysis of the DWHS and the two-stage training method of our framework is described in Section 6.3 and Section 6.4, respectively.

*6.1.2 Results on Single-labeled Datasets.* To further validate the effectiveness of MAJL in enhancing both tasks, we conduct an evaluation on the test sets of single-labeled datasets. The model under evaluation in this experiment is the same as the one discussed in Section 6.1.1. In the Stage II of training MAJL, additional single-labeled music data from both MUSDB18 and MIR_ST500 is utilized. Additionally, the test sets of the MSS and PE tasks are derived from MUSDB18 and MIR_ST500, respectively. The results on MUSDB18 and MIR_ST500 are summarized in Table 3 and Table 4, respectively.

Table 3: Performance comparison for the music source separation task on the test set of MUSDB18.

| Methods | SDR | GNSDR |
| --- | --- | --- |
| U-Net [29] | 6.72 | 13.38 |
| HT Demucs [53] | 7.93 | 14.58 |
| Hybrid Demucs [9] | 8.13 | 14.96 |
| CWS-PResUNet [40] | 8.92 | 15.49 |
| ResUNetDecouple+ [36] | 8.96 | 15.59 |
| KUIELAB-MDX-Net [33] | 9.00 | 15.64 |
| MAJL (Stage I with MedleyDB) | 9.46 | 15.94 |
| MAJL (Stage I with MIR-1K) | **10.13** | **16.51** |

The results indicate that our framework effectively learns the relationship between both tasks, enhancing the performance of each individual task. The results on the MUSDB18 dataset, presented in Table 3, show that our framework achieves state-of-the-art performance for the MSS task. Specifically, MAJL trained on the MIR-1K dataset in the Stage I outperforms the previous best MSS model

**Table 4: Performance comparison for the pitch estimation from mixture music on the test set of MIR_ST500.**

| Methods | RPA(%) | RCA(%) |
|---|---|---|
| CNN-Raw [11] | 76.58 | 77.26 |
| JDC [37] | 80.09 | 80.38 |
| U-Net [29] with CREPE [32] | 81.61 | 82.36 |
| ResUNetDecouple+ [36] with CREPE [32] | 81.96 | 82.43 |
| U-Net [29] with HARMOF0 [61] | 82.00 | 82.35 |
| ResUNetDecouple+ [36] with HARMOF0 [61] | 82.78 | 82.99 |
| MAJL (Stage I with MedleyDB) | 83.44 | 83.91 |
| MAJL (Stage I with MIR-1K) | **84.49** | **85.26** |

(KUIELAB-MDX-Net) by 1.13 in SDR. This result illustrates our framework has the capability to effectively leverage the PE task to improve the performance of the MSS task. Similarly, the results on the MIR_ST500 dataset, as shown in Table 4, demonstrate that our framework achieves state-of-the-art performance for the pitch estimation from mixture music. In particular, MAJL trained on the MIR-1K dataset in the Stage I outperforms the previous best method (ResUNetDecouple+ with HARMOF0) by 1.71% in RPA. This result indicates that our framework can effectively leverage the MSS task to enhance the performance of the PE task. In summary, these results highlight that our framework effectively leverages the mutually beneficial relationship between both tasks, thereby improving their individual performances.

## 6.2 Experiments With Different Modules

In this experiment, we investigate the effect of using different modules in our framework to demonstrate its generality. Our framework can employ various model architectures for the MSS Module and the PE Module. Therefore, we evaluate the performance of our framework using different combinations of these modules, including ResUNetDecouple+ [36], U-Net [29], HARMOF0 [61] and CREPE [32]. For this experiment, we utilize the MIR-1K dataset in Stage I because MAJL-Stage I, trained with the MIR-1K dataset, demonstrates superior performance on the MUSDB18 and MIR_ST500 datasets, thereby enabling the generation of better pseudo-labels. The results obtained with different combinations of these MSS and PE modules are summarized in Table 5. According to the results in this table, we can find three main observations as follows.

Firstly, this experiment shows the great generality of our framework. As shown in Table 5, MAJL consistently outperforms both the corresponding pipeline and naive joint learning methods across all module combinations. Particularly, when utilizing U-Net and CREPE, our framework achieves an improvement of 0.94 in SDR and 2.95% in RPA compared to the corresponding pipeline method. Similarly, our framework outperforms the corresponding naive joint learning method by 0.59 in SDR and 2.54% in RPA. These results validate the model-agnostic nature of our framework, showing its consistent performance enhancement across various music source separation and pitch estimation models.

Secondly, the two-stage training method and DWHS are robust with different MSS and PE modules. Specifically, MAJL using ResUNetDecouple+ and CREPE achieves the best performance. Notably, compared to MAJL-Stage I, MAJL achieves improvements of 0.65 in SDR and 0.94% in RPA, demonstrating the effectiveness of the two-stage training method in leveraging large single-labeled

**Table 5: Performance with different combinations of modules on the MIR-1K dataset. U, R, H, C represents the model architecture U-Net [29], ResUNetDecouple+ [36], HARMOF0 [61] and CREPE [32], respectively. MAJL here uses both MIR_ST500 and MUSDB18 as single-labeled data.**

| MSS | PE | Joint Method | MSS | | PE(%) | |
|---|---|---|---|---|---|---|
| | | | SDR | GNSDR | RPA | RCA |
| U | H | Pipeline | 11.43 | 8.48 | 87.95 | 88.57 |
| | | Naive Joint Learning | 11.05 | 8.11 | 88.96 | 89.20 |
| | | MAJL-Stage I | 12.05 | 9.11 | 90.62 | 91.57 |
| | | MAJL | 12.25 | 9.29 | 91.09 | 91.85 |
| R | H | Pipeline | 12.06 | 9.13 | 90.21 | 90.61 |
| | | Naive Joint Learning | 12.04 | 9.08 | 91.46 | 91.61 |
| | | MAJL-Stage I | 12.28 | 9.33 | 92.16 | 92.76 |
| | | MAJL | 12.60 | 9.64 | 92.51 | 93.16 |
| U | C | Pipeline | 11.43 | 8.48 | 89.28 | 90.41 |
| | | Naive Joint Learning | 11.78 | 8.84 | 89.69 | 90.44 |
| | | MAJL-Stage I | 12.16 | 9.19 | 91.78 | 92.40 |
| | | MAJL | 12.37 | 9.41 | 92.23 | 92.79 |
| R | C | Pipeline | 12.06 | 9.13 | 91.40 | 92.07 |
| | | Naive Joint Learning | 11.91 | 8.92 | 91.88 | 92.15 |
| | | MAJL-Stage I | 12.33 | 9.36 | 93.17 | 93.65 |
| | | MAJL | **12.98** | **9.99** | **94.11** | **94.38** |

music data. Moreover, MAJL-Stage I outperforms the naive joint learning method by 0.42 in SDR and 1.29% in RPA, indicating that the DWHS effectively aligns different objectives and enhances the performance of both tasks. Furthermore, similar performance trends are observed with MAJL and MAJL-Stage I utilizing alternative MSS and PE modules. These results demonstrate that both the two-stage training method and the DWHS are model-agnostic and robust, further confirming the model-agnostic nature of our framework.

Lastly, our framework effectively learn the relationship between MSS and PE tasks, making both tasks beneficial for the each other. For example, MAJL using ResUNetDecouple+ and CREPE outperforms MAJL using U-Net and CREPE by 0.61 in SDR and 1.88% in RPA. This improvement arises from the better performance of ResUNetDecouple+ in the MSS task, where the effectiveness of the MSS task is beneficial for the PE task. Similarly, MAJL using ResUNetDecouple+ and CREPE outperforms MAJL using ResUNetDecouple+ and HARMOF0 by 0.38 in SDR and 1.60% in RPA. This is because CREPE performs better than HARMOF0 at the PE task, and the PE task is beneficial for the MSS task. Thus, our framework also has the potential to further improve the performance of both tasks when better MSS and PE models become available.

The above results demonstrate the robustness and great generality of our framework, since our framework achieves the best performance across all module combinations. Moreover, our MAJL framework effectively learns the relationship between MSS and PE tasks, resulting in enhanced performance for both tasks.

## 6.3 Visualization and Analysis of Dynamic Weights

To provide an intuitive representation of the weights extracted by the DWHS, we visualize the changes in these weights over iterations. In this experiment, we employ ResUNetDecouple+ as the MSS Module and CREPE as the PE Module, consistent with the previous experiment detailed in Section 6.1. In addition, the dynamic weights extracted by Dynamic Weights on Hard Samples (DWHS)

are obtained from our framework, specifically MAJL-Stage I, to exclusively investigate the weight results of DWHS and eliminate potential interference, such as single-labeled music data.

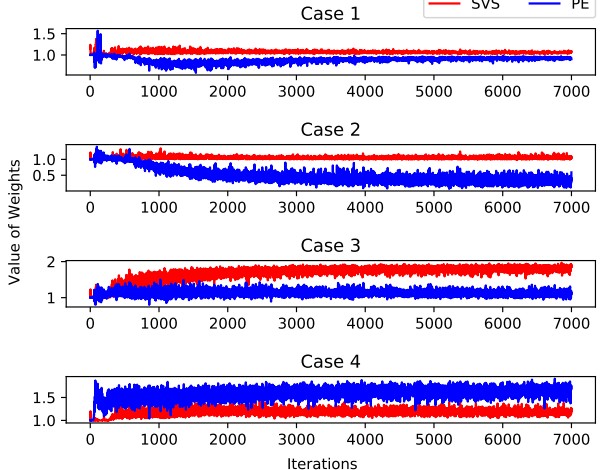

Figure 5: Dynamic weights extracted by the DWHS.

Figure 5 illustrates the dynamic weights extracted by the DWHS. In this figure, we observe that the weight assigned to the PE task for noisy music data is set close to 0, effectively mitigating the negative impact on the PE task. Additionally, for Case 3 and Case 4, the weights assigned to MSS and PE exceed 1, emphasizing the importance of handling hard samples. Other weights are approximately 1. All these weights are automatically set based on the analysis presented in Table 1. Furthermore, the results presented in Table 2 demonstrate that the DWHS method outperforms the corresponding naive joint learning method. These findings highlight that the DWHS method can adaptively determine appropriate weights for both noisy and hard samples, leading to enhanced performance in both MSS and PE tasks.

## 6.4 Threshold in Two-Stage Training Method

In this experiment, we use the MUSDB18 and MIR_ST500 datasets as single-labeled music data to explore the impact of the threshold ($th$) used in the two-stage training method. The ResUNetDecouple+ is used as the MSS Module and CREPE is used as the PE Module, consistent with the previous experiment in Section 6.1. The MAJL is initially trained on the MIR-1K dataset in Stage I, as it demonstrates superior performance compared to MAJL trained on the MedleyDB dataset, as evidenced by the results presented in Table 3 and Table 4.

As shown in Figure 6, the results corresponding to different thresholds demonstrate the effectiveness of the two-stage training method, and that the chosen threshold influences the performance of MSS and PE tasks. Specifically, when incorporating both datasets (MUSDB18 and MIR_ST500) to the fully-labeled music data, the performance of both tasks firstly improved and then decreased as the threshold increased. This trend indicates that the quality of pseudo-labels affects the performance of both tasks, with lower-quality pseudo-labels leading to a decrease in the performance of both tasks. In addition, incorporating the MUSDB18 or MIR_ST500 dataset to the fully-labeled music data follows a similar trend in the performance of both tasks. Moreover, when the threshold was set

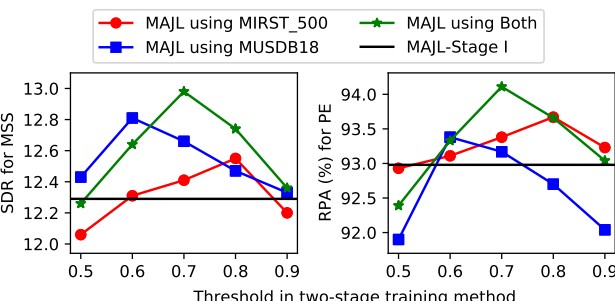

Figure 6: Performance with different values of threshold ($th$) in two-stage training method on MIR-1K dataset. MAJL-Stage I represents Stage I in the two-stage training method.

to 0.7, adding both datasets (MUSDB18 and MIR_ST500) to the fully-labeled music data achieves the best performance, outperforming MAJL-Stage I by 0.65 in SDR and 0.94% in RPA. These results show that the two-stage training method can leverage large single-labeled music data to enhance the performance of both tasks.

## 7 FUTURE WORK

Our experiments have primarily focused on vocals, given that vocals are a common target source for both MSS and PE tasks, as highlighted in previous studies [28, 44]. Furthermore, the availability of music data containing other target sources and corresponding pitches was limited. However, it is important to note that our framework is applicable to a variety of musical instruments, including drums, bass and so on.

Expanding our framework to encompass other instruments requires the acquisition or creation of annotated data for both MSS and PE tasks. Overcoming this challenge may involve adapting transfer learning techniques from related domains or exploring further unsupervised training methods. In conclusion, the future of our research holds the potential to adapt and expand our Model-Agnostic Joint Learning (MAJL) framework to encompass a broader range of musical instruments. These directions align with the ongoing evolution of music production techniques and computational audio analysis, making our framework important in advancing the field of music information retrieval.

## 8 CONCLUSION

In this paper, we have proposed a model-agnostic joint learning (MAJL) framework to address the challenges in joint learning of music source separation and pitch estimation. MAJL is generic for both tasks in music information retrieval, offering the capability to adapt improved models for both tasks, further improving their performance. By designing a two-stage training method and a dynamic weighting method named *Dynamic Weights on Hard Samples* (DWHS), our framework effectively addresses the challenges of the lack of labeled data and the joint learning optimization, respectively. Through leveraging extensive single-labeled music data, MAJL learns the mutually beneficial relationship between music source separation and pitch estimation tasks, leading to improved performance for both tasks. Our experimental results show that the proposed framework achieves a significant improvement in both tasks, with 0.92 in SDR for the music source separation task and 2.71% in RPA for the pitch estimation task.

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
