# OpenReview forum: "MAJL: A Model-Agnostic Joint Learning Framework for Music Source Separation and Pitch Estimation"
_acmmm.org/ACMMM/2024/Conference — MM2024 Poster_

### Official Review · Reviewer_vUBJ · 2024-05-02

**Rating:** 4
**Confidence:** 2

**Summary:**

This paper proposes a model-agnostic framework for music source separation and pitch estimation. A two-stage training method and a dynamic weighting method are introduced to address the lack of labeled data and the joint learning optimization issue. Experimental results show that the framework achieves a significant improvement in both tasks.

**Strengths:**

1. This paper is well-written and the motivation and description of the proposed method are very clear.
2. The experiment is sufficient and demonstrates the effectiveness of proposed framework in source separation and pitch estimate tasks.
3. The model-agnostic joint learning framework can learn the mutually beneficial relationship between music source separation and pitch estimation tasks by leveraging extensive single-labeled music data

**Limitations:**

This paper lacks of novelty, the model agnostic framework mainly relies on joint training which is not a very novel method.

**Suitability:**

2

---

### Official Review · Reviewer_92gM · 2024-05-06

**Rating:** 5
**Confidence:** 2

**Summary:**

This paper introduces a model-agnostic joint learning (MAJL) framework aimed at tackling the inherent challenges in simultaneously learning music source separation and pitch estimation. MAJL offers a versatile solution applicable to both tasks, enabling the incorporation of enhanced models tailored to each task. The implementation of a two-stage training approach and a novel dynamic weighting mechanism termed Dynamic Weights on Hard Samples (DWHS) effectively address the challenges of the lack of labeled data and the joint learning optimization.

**Strengths:**

1. The paper is well written, well organized and provides a good rationale for their motivation with a detailed literature review.
2. This paper clearly clarifies two challenges when performing MSS and PE tasks simultaneously.
3. The proposed MAJL framework simultaneously tackle MSS and PE tasks and achieves the state-of-the-art performance.
4. The experiments were adequately designed.

**Limitations:**

1. Typo: “SVS” in the legend of Figure 5.

**Suitability:**

2

---

### Official Review · Reviewer_qC7o · 2024-05-23

**Rating:** 3
**Confidence:** 3

**Summary:**

To address the challenges of the lack of the labeled data and join learning optimization in music source separation and pitch estimation, the authors proposed a MAJL framework which includes a two-stage training method and a dynamic weighting method (DWHS). The two-stage training method effectively leverages extensive single-labeled datasets and high-quality pseudo labels. The DWHS can handle error propagation and misalignment between different objectives by identifying hard samples and setting appropriate weights for these samples. Experimental results on public music datasets show that MAJL outperforms state-of-the-art methods on both tasks.

**Strengths:**

- The use of single-labeled datasets and the dynamic weighting method for hard samples based on several cases are highly novel. The loss function part of the dynamic weight module is particula interesting.
- It is good to show the effectiveness of the proposed method by comparing it with state-of-the-art methods in an exhaustive manner.
- The validity of the proposed framework is confirmed by checking the performance when using different model structures in MSS and PE.

**Limitations:**

- It was not clear why equation (13) added up the loss functions for all cases. How is equation (14) optimized for each case?
- It is unfortunate that the experiment is limited to vocal parts. If the authors claim this method is for music source separation, I would have liked to have seen experimental results showing that it can be used to separate other musical instruments as well.
- It would be good to have a discussion on the relationship between data volume in single-labeled datasets and fully-labeled datasets and performance.
- This research is limited to the modality of music and cannot be called a multimodal research.

**Suitability:**

1

---

### Official Review · Reviewer_kfcD · 2024-05-24

**Rating:** 2
**Confidence:** 4

**Summary:**

This paper proposed a model-agnostic joint learning for music source separation and pitch estimation, which is interesting for the audience of ACM Multimedia. However, there are several concerns make me believe that this paper is not ready for publication. The detailed comments can be found below.

**Strengths:**

The writing of this paper is good. The presentation of this paper is clear.

**Limitations:**

i) The most important issue of this paper is a lack of basic background knowledge of music information retrieval. The concept of pitch estimation (PE) and melody extraction is not clear in this paper. Pitch estimation commonly predicts pitch contours of each part of the polyphonic music and singing melody extraction predicts the fundamental frequency of the music. However, the paper uses pitch estimation algorithm to estimate fundamental frequency. Thus, actually the paper is doing singing melody extraction. It can be confirmed from Table 2 and Table 4, only fundamental frequencies are predicted.

ii) Moreover, many important related works are missing. For example, other than HT Demucs, the rest of baseline methods are published before the year of 2022. There are many newly published music source separation methods [a,b] are worthy to mention and compare with. Since this paper is actually doing singing melody extraction, many singing melody extraction methods [c,d] are not cited and compared.

iii) The novelty of this paper is limited, as semi-supervised learning for melody extraction has been proposed [d,e].

[a] Schulze-Forster, Kilian, et al. "Unsupervised music source separation using differentiable parametric source models." IEEE/ACM Transactions on Audio, Speech, and Language Processing 31 (2023): 1276-1289.

[b] Luo, Yi, and Jianwei Yu. "Music source separation with band-split RNN." IEEE/ACM Transactions on Audio, Speech, and Language Processing (2023).

[c] Gao, Yuan, et al. "MTANet: Multi-band time-frequency attention network for singing melody extraction from polyphonic music." Proc. INTERSPEECH 2023. 2023.

[d] Yu, Shuai. "MCSSME: Multi-Task Contrastive Learning for Semi-supervised Singing Melody Extraction from Polyphonic Music." In Proceedings of the AAAI Conference on Artificial Intelligence, vol. 38, no. 1, pp. 365-373. 2024.

[e] Kum, Sangeun, Jing-Hua Lin, Li Su, and Juhan Nam. "Semi-supervised learning using teacher-student models for vocal melody extraction." arXiv preprint arXiv:2008.06358 (2020).

**Suitability:**

2

---

### Meta-Review · Program_Chairs · 2024-07-01

**Recommendation:** Accept (Poster)
**Confidence:** 4

**Metareview:**

This paper proposes a model-agnostic framework for music source separation and pitch estimation. It includes a two-stage training method and a dynamic weighting method to address the lack of labeled data and the joint learning optimization issue. Experimental results show that the framework achieves a significant improvement in both tasks. All reviewers appreciate the novel contributions of the paper, and acknowledge that this paper is well-written. However, all reviewers agree that this research is single modal, not multimodal research. This is evidenced by the fact that the vast majority of the references are from the signal processing communities, including ICASSP, TASLP and ISMIR. This paper will fit beautifully to above venues. I am puzzled why the authors have decided to submit the paper to MM. In consideration of all review comments as well as the authors' rebuttal, I am unable to recommend an accept, unfortunately. However, I would strongly encourage the authors to take the review comments into consideration, and to improve the manuscript for a more suitable venue such as TASLP.

*** TPC Addendum ***
The paper has received positive scores and they have gotten better with the rebuttal. While the work is not completely multi-modal it does have significant overlaps with research interests of the multimedia community. Hence, the TPC is recommending a poster accept to allow for the conversation to continue at the conference.